# RGB-D-Based Pose Estimation of Workpieces with Semantic Segmentation and Point Cloud Registration

**DOI:** 10.3390/s19081873

**Published:** 2019-04-19

**Authors:** Hui Xu, Guodong Chen, Zhenhua Wang, Lining Sun, Fan Su

**Affiliations:** School of Mechanical and Electric Engineering, Jiangsu Provincial Key Laboratory of Advanced Robotics, Collaborative Innovation Center of Suzhou Nano Science and Technology, Soochow University, Suzhou 215123, China; yanwu19@126.com (H.X.); wangzhenhua@suda.edu.cn (Z.W.); lnsun@hit.edu.cn (L.S.); fsu@stu.suda.edu.cn (F.S.)

**Keywords:** RGB-D, industrial scenarios, pose estimation, semantic segmentation, homemade dataset, point cloud registration, robot vision system

## Abstract

As an important part of a factory’s automated production line, industrial robots can perform a variety of tasks by integrating external sensors. Among these tasks, grasping scattered workpieces on the industrial assembly line has always been a prominent and difficult point in robot manipulation research. By using RGB-D (color and depth) information, we propose an efficient and practical solution that fuses the approaches of semantic segmentation and point cloud registration to perform object recognition and pose estimation. Different from objects in an indoor environment, the characteristics of the workpiece are relatively simple; thus, we create and label an RGB image dataset from a variety of industrial scenarios and train the modified FCN (Fully Convolutional Network) on a homemade dataset to infer the semantic segmentation results of the input images. Then, we determine the point cloud of the workpieces by incorporating the depth information to estimate the real-time pose of the workpieces. To evaluate the accuracy of the solution, we propose a novel pose error evaluation method based on the robot vision system. This method does not rely on expensive measuring equipment and can also obtain accurate evaluation results. In an industrial scenario, our solution has a rotation error less than two degrees and a translation error < 10 mm.

## 1. Introduction

Industrial robots have been widely deployed in the manufacturing sector, especially in the production of high volume products. Industrial robots, with their stable performance and long-term operation, are generally more advantageous than workers are for a wide range of repetitive tasks. To reduce production and labor costs, an increasing number of enterprises are hoping that industrial robots can undertake more loading and unloading work in the front and rear processes, such as removing the front-end production line for sorting scattered workpieces and directly dispersing the workpieces simultaneously using industrial robots. Traditional industrial robots that are skilled in repetitive operations have been unable to meet the requirements of these new application scenarios. Subsequently, intelligentization has become the key to solving these problems.

As for the eyes of industrial robots, vision can accomplish image classification, recognition, and detection tasks excellently [1], and is an important part of industrial robot intelligence. To grasp scattered workpieces in industrial scenarios, industrial robots need to acquire six degrees of freedom information (translation x, y, z, and rotation A, B, C) in 3D (three-dimensional) space in order to determine the pose of the workpiece. However, 2D (two-dimensional) color images usually only provide two or three pieces of information (x, y or x, y, A). To solve this problem of insufficient space pose information of the workpiece, one method is to collect the depth information corresponding to the color image synchronously and then construct and process the 3D point cloud model of the working scene by using the depth information to obtain the pose of the workpiece.

Pose estimation and grasping strategy are two important research topics for intelligently grasping scattered workpiece tasks, which have become the frontier of industrial robot operation research. For the input of the subsequent grasping strategy research, pose estimation is the key technology of the entire grasping system. To estimate the pose of objects, several researchers recently focused on training RGB images end-to-end directly to output pose estimation results by using CNNs (convolutional neural networks) [2,3], such as BB8 [4], SSD-6D [5], PoseCNN [6], and YOLO-6D [7]. Among these methods, BB8 and PoseCNN have similar architectures that employ multiple stages to refine ambiguous estimates of the 2D locations of projected vertices of each object’s 3D bounding cuboid, then 6D poses are obtained by using the PnP algorithm. Both BB8 and SSD-6D require a further pose refinement step for improved accuracy, which increases their running times linearly with the number of objects being detected. To address this problem, YOLO-6D takes the image as input and directly detects the 2D projections of the 3D bounding box vertices, which is end-to-end trainable without any a posteriori refinement. However, these works require a complicated labeled dataset, such as LINEMOD [8] or OCCLUSION [9], for training, which is difficult to make by ourselves.

With the popularity of RGB-D (color and depth) sensors, many novel and practical methods for object pose estimation have been recently proposed [10,11,12,13]. Wong et al. proposed a perception framework that combines deep semantic segmentation and model-based object pose estimation, which uses RGB-D sensors to provide semantic segmentation of all relevant objects in the scene along with their respective poses [14]. Zeng et al. used the FCN to perform semantic segmentation from the RGB-D image and then estimated its 3D pose by aligning the segmented point clouds with an original 3D model [15]. Lin proposed a system that fuses the deep-learning based semantic segmentation method and a RANSAC-based pose estimation method to process the task of object recognition and pick pose estimation [10]. These methods generally include two components: (1) a deep CNN-based semantic segmentation method is used to simultaneously detect and recognize multiple objects, and (2) a pose estimation method is utilized to perform pose estimation [16].

According to the method mentioned above, we focus on the relevant studies related to object segmentation and pose estimation in this work.

There are currently two primary approaches for object segmentation in the point cloud of a scattered industrial setting. The first is to use the geometric features of the surface of the point cloud, such as normal, concavity, connectivity, etc., to directly segment the 3D point cloud [17,18,19]. It is easy for such methods to introduce a large number of scattered points, and their performances are greatly affected by the clutter scene. The second is to identify and segment the objects in the RGB images and then map out the point cloud of the objects based on the characteristics of the RGB-D data [20]. Such methods are mainly affected by the precision of the color map semantic segmentation. Semantic segmentation performs a pixel-wise classification to mark the position of the object at the pixel level. Therefore, semantic segmentation can provide more specific object information. In recent years, deep learning has achieved excellent results in the fields of object classification and detection [21,22]. Ciresan et al. attempted to apply a CNN to image semantic segmentation, training a CNN model to classify each pixel based on its patch category in the image [23]. However, the segmentation speed of this method is relatively slow. Based on the CNN architecture, Long et al. proposed an FCN (Fully Convolutional Network), which creatively replaces the fully connected layers in the network with the convolutional layers to implement an end-to-end pixelwise semantic segmentation prediction [24]. In both CNN and FCN networks, the pooling layer can expand the receptive fields of the neural network while lowering the resolution of the image [25]. Yu et al. proposed the context module, which uses dilated convolutions for multi-scale aggregation, in which the dilated convolution layer can greatly expand the receptive field without reducing the spatial dimension [26]. To obtain more refined segmentation results, Chen et al. presented atrous spatial pyramid pooling (ASPP) to fuse multiscale information. The segmented image is subsequently refined by a fully connected CRF, which is trained separately as a post-processing step [27,28].

To address the problem of estimating the free-form 3D objects’ poses in point clouds, Drost et al. created a global model description and locally matched the model by using a fast voting scheme [29]. The global model in this method is described based on oriented point pair features. In theory, the relative position of the two point clouds can be addressed by three point pairs, but in the process of processing the point cloud, it is difficult to obtain exact corresponding point pair features. Many researchers have proposed more significant local feature descriptors (SHOT, SIFT et al.), which are mainly used on highly textured and rigid objects to estimate the pose from the local and global pipelines [30,31,32]. For texture-less objects captured by an RGB-D sensor, Hinterstoisser et al. proposed a template-based LINEMOD approach that matches the objects to the template model in the library using the color gradients of the RGB image and the normal features of the depth image [8,33]. However, the results obtained directly from the template matching are rough and not exact, thus we usually use the ICP (Iterative Closest Point) for pose refinement. Based on the Euclidean distance minimum criterion, the ICP algorithm, which was developed by Besl and McKay to iteratively search the corresponding points between the source model and the target model [34], has been one of the most popular registration methods for unorganized point cloud datasets to date. The advantage of this algorithm is that it does not need to be specified of any corresponding features in advance, while the drawback is that it may result in the iterative process converging to a local minimum when the initialization information is insufficient, such as when there are few overlapping regions between the two point clouds. Therefore, some researchers have proposed a variety of registration preprocessing methods based on the ICP algorithm [14,16]. For example, Rusu proposed the FPFH-ICP registration algorithm. He first established the point feature histogram by using the normal features of the point cloud and then integrated the sample consensus algorithm to achieve the initial alignment of the two point clouds. He then initialized the input transformation matrix of the ICP algorithm with the preprocessed result of the initial alignment to avoid falling into the local minima [35]. To achieve higher robustness against the possible scenario changes, Marani et al. proposed a modified ICP algorithm by introducing deletion masks that can remove the measurement deviation generated by the changes of the camera’s viewpoint [36]. Yang et al. presented a globally optimal solution to 3D Euclidean registration with a trimming strategy to handle outliers of the point cloud, called Go-ICP. The method is based on the Branch-and-Bound (BnB) algorithm that can guarantee global optimality even if good initialization is not reliably available, but real-time performance of the algorithm needs to be further enhanced.

Therefore, in order to grasp the scattered workpieces, this work introduces a sensor (including a color camera and a depth camera) that can simultaneously collect all RGB-D data for an industrial robot system. Then, we propose a method to identify and estimate the pose of the workpiece in 3D space using the RGB-D data. The method consists of three phases: first, we construct a full convolutional neural network for image semantic segmentation, and then use transfer learning technology to train this network on a homemade dataset; second, we use the trained full convolutional neural network model to segment the scattered artifacts from the RGB images captured by the color camera, and then combine the depth information to map the corresponding workpiece point cloud model; finally, based on the point cloud registration method, we align the mapped point cloud with the initial point cloud of the workpiece to estimate the transformation matrix between the two point clouds. After a series of matrix transformations, we finally obtain the pose of the workpiece in the industrial robot base frame.

## 2. Methods

In order to reduce the investment of personnel and equipment, some enterprises are introducing industrial robots to perform front-end grasping operations. As shown in Figure 1a, we built a workpiece grasping experiment platform to simulate the workpiece grasping process in the industrial scenario. In this work, the target objects are the shells of the industrial robot, as shown in Figure 1b. The Workpiece-01 and the Workpiece-02 are the industrial robot’s shells, which have been sprayed and will be assembled on an industrial robot. The Workpiece-03 and the Workpiece-04 are semi-finished products, which have not been sprayed and are used in industrial robot experiments and teaching activities. The RGB-D (Kinect V2) camera first acquires the color and depth information of the industrial robot, the target object, and the surrounding environment. By processing this information, the computer transmits the pose data of the workpieces to the industrial robot. Then, the industrial robot sequentially grasps the workpieces based on the pose information relative to the initial pose, and the grasping mode of the workpieces in the initial pose is set by manual teaching.

Based on the novel methods proposed by several researchers, we propose a feasible pose estimation method for the workpieces with characteristics of the industrial scenario, as shown in Figure 2.

### 2.1. Object Segmentation Based on Modified FCN

The RGB-D sensor can capture both RGB and depth images synchronously, where the RGB image contains much of the object information. In recent years, CNNs have made great progress in processing RGB images, especially in difficult and complex object detection fields, where the CNN can quickly and accurately draw the bounding box around the target object [37]. However, the precision of the CNN-based method for image semantic segmentation is low and hard to improve compared to CNN-based classification, localization, and detection. One of the reasons for this is that the patchwise training in CNN-based segmentation does not take the correlation between image patches into account and therefore loses the spatial information. Therefore, Long et al. re-architected and fine-tuned CNN-based classification networks for the direct, dense prediction of end-to-end, pixel-to-pixel semantic segmentation [24]. CNN networks such as AlexNet, VGGNet, GoogleNet and ResNet usually contain convolutional layers, pooling layers, and fully connected layers. The convolutional layers are the core of the deep neural network, which are used to extract color, texture, shape, and other features of the images. The pooling layers are used to extract abstract features and reduce the complexity of the subsequent convolution calculations, as well as prevent overfitting. The fully connected layers are used to classify the extracted features, but the classification result does not contain the spatial information of the pixels, nor can it be used for the image semantic segmentation tasks. Therefore, the FCN inherits the convolutional layers and the pooling layers of the CNN front architecture, extracts the useful feature information from the image, and then replaces the fully connected layers with the convolutional layers to output a heatmap containing the spatial information. On one hand, the pooling layers reduce the size of the output image, thereby significantly reducing the amount of computation. On the other hand, the pooling layer increases the receptive field while losing the position information of many pixels, which results in the resolution size of the output heatmap being reduced by several times compared to the original image, thus it cannot be used directly for semantic segmentation. To restore the heatmap to the same resolution of the input image and refine the segmented image, we upsample the final output heatmap stage-by-stage using multiple skip architectures to fuse the substantial feature information in the front pooling layers, thus each pixel with spatial information in the final output image has a corresponding prediction value that indicates the most likely class it belongs to.

Following this method, we cast a VGG architecture, an effective and efficient convolutional neural network architecture that won the ILSVRC14 image classification challenge [1,22], into the FCN-atrous-2s to perform 2D object segmentation, as shown in Figure 3. In this work, we mainly perform research on object segmentation approaches for automation industrial workpieces. These workpieces differ in color, texture, shape, and other characteristics from the PASCAL VOC 2012 dataset used to train the FCN [38]. To accurately and completely acquire the pose of the target workpieces, we adapt the VGG-based FCN architectures by fine-tuning the fully convolution layers and adding more skips between layers. We also retrain the FCN-atrous-2s end-to-end from the whole image inputs and whole image labels by using the supervised classification pre-training model VGG16.

### 2.2. Pose Estimation Based on ICP

Semantic segmentation can identify, locate, and segment objects in a 2D RGB image. However, the pose of an object is represented by six degrees of freedom in 3D space. Therefore, the objective of pose estimation is to calculate the most likely 6-DOF pose of the object, which is the closest pose to the true value. The 3D point cloud of an object can be mapped from the segmentation result inferred by the FCN-atrous-2s and the corresponding depth image. A variety of point cloud registration methods have been proposed to address the pose estimation. Point cloud registration is a technique for aligning and splicing point clouds (with overlapping regions between them) of the same object at multiple viewpoints into a complete model. The core of this technique is to calculate the relative poses between the point clouds in the global coordinate system [35]. Therefore, point cloud registration can be used not only for splicing point clouds but also for calculating the relative poses between the point clouds. As shown in Figure 4, we register the segmented point cloud with the initial point cloud to obtain the relative pose of the target workpiece by using point cloud registration technology in this work.

#### 2.2.1. Point Cloud Preprocess

Due to the influence of equipment accuracy, surface properties of the object, and environmental factors, the point clouds captured and generated using the RGB-D sensor inevitably contain some noise and scattered points that are far from the main point clouds. Moreover, the amount of data captured directly by the camera is usually very large and, to a certain extent, redundant, which seriously affects the speed and accuracy of the subsequent point cloud processing, which is not suitable for tasks (such as robot grasping) with high real-time requirements. Therefore, in the preprocessing phase, it is very necessary to use a filtering algorithm to remove outliers and downsample the point cloud. We use the statistical outlier removal method to remove the noise and outliers [39]. Based on the distance distribution from each point to the adjacent points, we perform a statistical analysis on the neighborhood of each point and trim out the points that do not meet the criteria. For each point $p_{q}$ in the point cloud, we calculate its mean distance $\bar{d}$ to the K nearest neighbors. It is assumed that the mean distance result of the entire point cloud is a Gaussian distribution whose shape is determined by the mean $\mu_{k}$ and the standard deviation $\sigma_{k}$. The points whose mean distance do not meet the requirements are defined as outliers and are removed from the point cloud. The remaining points $p_{q}^{*}$ can be represented by Equation (1).
(1)$$P^{*}=\left\{p_{q}^{*}\in P|\left(\mu_{k}-\sigma_{k})\leq \bar{d}\leq (\mu_{k}+\sigma_{k}\right)\right\}$$

The amount of point cloud data is still enormous after the outliers are removed, thus we use the VoxelGrid method for point cloud downsampling. The VoxelGrid can reduce the number of points while keeping the contour of the point cloud unchanged, and it can also improve the running speed of the algorithms, such as point cloud feature recognition and registration algorithms. VoxelGrid creates multiple 3D voxels (i.e., tiny spatial 3D cubes) in the point cloud based on the input size parameters *Ls*. Each voxel is a small point set containing a different number of points. We then calculate the centroid of all points in each voxel through Equation (2), and use this centroid to approximately represent all the points in the corresponding voxel. The effect of the VoxelGrid downsampling is very significant, as it can generally reduce the number of points above an order of magnitude in voxels with a high point cloud density. Therefore, the downsampling method is an indispensable preprocessing phase in scenarios where real-time requirements are very high.
(2)$$\mu_{P}=\frac{1}{m}\sum_{i=1}^{m}p_{i}$$
where $p_{i}$ and *m* are the points contained in one voxel and the number of the points, respectively.

As shown in Figure 2, when the target workpiece types in the scene are different, the output image of semantic segmentation will be rendered with different colors for each distinct workpiece. However, when there are more than one workpieces that belong to the same type in the scene, the same workpieces are rendered with the same color in the semantic segmentation image. In the process of generating a point cloud, it is hard to divide the same type of workpieces into multiple single point clouds, which affects subsequent point cloud registration operations. Instance segmentation technology, a combination of semantic segmentation and object detection, can be used to address this problem [40,41]. However, in this work, we focus on the study of semantic segmentation methods and extract the distinct point clouds by utilizing Euclidean clusters method [35]. The method can be described as:(3)$$\min{\left\|p_{i}-p_{j}\right\|}_{2}\geq d_{th}$$
where points sets *p_i_* and *p_j_* belong to the different point clusters, and *d_th_* is a maximum imposed distance threshold. The above equation represents that the minimum distance between *p_i_* and *p_j_* should be larger than a given distance value, otherwise the points in *p_i_* and *p_j_* are identified as the same cluster.

#### 2.2.2. Principle of ICP method

The process of point cloud registration involves searching for the optimal transformation matrix to minimize the sum of the Euclidean distances between the corresponding points of the source point cloud P transformed by the optimal transformation matrix and the target point cloud V, as described in Equation (4):(4)$$f=\min\left(\sum_{i=0}^{m}\sqrt{{\left(x_{i}^{P}-x_{i}^{V}\right)}^{2}+{\left(y_{i}^{P}-y_{i}^{V}\right)}^{2}+{\left(z_{i}^{P}-z_{i}^{V}\right)}^{2}}\right)$$

However, for the two sets of point clouds captured from the different viewpoints, it is difficult to guarantee that the points of the overlapping regions of the two viewpoints have a one-to-one correspondence, which leads to the direct calculation of Equation (4) being theoretically feasible but difficult to implement in practical applications. Therefore, we introduce the ICP algorithm in this work. This algorithm first randomly generates the initial transformation matrix $\left(R_{init},q_{init}\right)$, then it determines the transformation matrix $T_{reg}$ that satisfies the convergence condition through multiple iterations and updates. Assuming that $p_{i}$ and $v_{j}$ are arbitrary points in the target point cloud and the source point cloud, respectively, the Euclidean distance between the two points can be expressed as:(5)$$d(p_{i},v_{j})=\left\|p_{i}-v_{j}\right\|=\sqrt{{\left(x_{i}-x_{j}\right)}^{2}+{\left(y_{i}-y_{j}\right)}^{2}+{\left(z_{i}-z_{j}\right)}^{2}}$$
where $p_{i}={\left(x_{i},y_{i},z_{i}\right)}^{T}$ and $v_{j}={\left(x_{j},y_{j},z_{j}\right)}^{T}$.

Then, the Euclidean distance between a point $v_{i}$ and point set *P* is defined by:(6)$$d(p_{i},V)=\underset{j\in \left\{1,2,\ldots ,n\right\}}{\min}d(p_{i},v_{j})=d(p_{i},v_{i}^{\prime })$$
where $v_{i}^{\prime }$ is the point in the target point cloud *V* that is closest to point $p_{i}$. By analogy, for each point in the source point cloud *P*, there is always a corresponding point in the point cloud *V* that is closest to it. These closest points compose a new point cloud $V^{\prime }=\left\{v_{i}^{\prime }|i=1,2,\ldots ,m\right\}$, which forms a new cross-covariance matrix with the source point cloud *V*:(7)$$\Sigma_{PV^{\prime }}=\frac{1}{m}\sum_{i=1}^{m}\left[\left(p_{i}-\mu_{P}\right){\left(v_{i}^{\prime }-\mu_{V^{\prime }}\right)}^{T}\right]=\frac{1}{m}\sum_{i=1}^{m}\left[p_{i}v_{i}^{\prime }{}^{T}\right]-\mu_{P}\mu_{V^{\prime }}{}^{T}$$
where $\mu_{P}$ and $\mu_{V^{\prime }}$ are the centroids of point clouds *P* and $V^{\prime }$, respectively, which can be calculated by Equation (2). According to the definition of the cross-covariance matrix, $M_{ij}={\left(\Sigma_{PV^{\prime }}-\Sigma_{PV^{\prime }}^{T}\right)}_{ij}$ are the cyclic elements of the anti-symmetric matrix. It is assumed that the column vector $A={\left[\begin{array}{ccc} M_{23} & M_{31} & M_{12} \end{array}\right]}^{T}$, a 4 × 4 symmetric matrix, can be derived based on this vector and Equation (7):(8)$$M\left(\Sigma_{PV^{\prime }}\right)=\left[\begin{array}{cc} tr\left(\Sigma_{PV^{\prime }}\right) & A^{T}\\ A & \Sigma_{PV^{\prime }}-\Sigma_{PV^{\prime }}{}^{T}-tr\left(\Sigma_{PV^{\prime }}\right)I_{3\times 3} \end{array}\right]$$

We use the SVD (singular value decomposition) method to solve the eigenvalues and eigenvectors of $M(\Sigma_{PV^{\prime }})$. The unit eigenvector $q_{R}={\left[\begin{array}{cccc} q_{w} & q_{x} & q_{y} & q_{z} \end{array}\right]}^{T}$, which corresponds to the maximum eigenvalue of the matrix $M\left(\Sigma_{PV^{\prime }}\right)$, is chosen as the optimal quaternion vector for rotation. Based on the Rodrigues formula, a unit quaternion can generate a 3 × 3 rotation matrix as follows:(9)$$R(q_{R})=\left[\begin{array}{ccc} q_{w}^{2}+q_{x}^{2}-q_{y}^{2}-q_{z}^{2} & 2(q_{x}q_{y}-q_{w}q_{z}) & 2(q_{x}q_{z}+q_{w}q_{y})\\ 2(q_{x}q_{y}+q_{w}q_{z}) & q_{w}^{2}-q_{x}^{2}+q_{y}^{2}-q_{z}^{2} & 2(q_{y}q_{z}-q_{w}q_{x})\\ 2(q_{x}q_{z}-q_{w}q_{y}) & 2(q_{y}q_{z}+q_{w}q_{x}) & q_{w}^{2}-q_{x}^{2}-q_{y}^{2}+q_{z}^{2} \end{array}\right]$$
where $R(q_{R})$ is the rotation component of the transformation matrix $T_{reg}$, and the corresponding optimal translation vector in the transformation matrix is described as:(10)$$q_{t}=\mu_{V^{\prime }}-R(q_{R})\mu_{P}$$

Thus, the mean square error between point cloud $V^{\prime }$ and point cloud *P* is as shown in Equation (11). We use this formula as the objective function: when *f* is less than the preset threshold, it is considered to satisfy the convergence condition.
(11)$$f=\frac{1}{m}\sum_{i=0}^{m}{\left\|v_{i}^{\prime }-R(q_{R})p_{i}-q_{t}\right\|}^{2}$$

What we describe above is an iterative process of ICP registration. A new point cloud $P^{\prime }$ is transformed from the source point cloud *P* by multiplying $T_{reg}$, which is determined through the ICP method during each iterative process. We then continue to register point cloud $P^{\prime }$ with the target point cloud *V* until the convergence condition is met.

## 3. Results and Discussion

### 3.1. Results of Semantic Segmentation

In industrial tasks, scenarios requiring robotic grasping operations are usually structured with fewer textures and a single background. Images for neural network training with a single background result in a trained neural network that may simply produce a “correct” inference based on the unrelated details of the training images. Even though the trained neural network can produce good results on the training set images, it still fails to perform a dense prediction on the new images out of the training set. This is mainly because the neural network model does not learn the general characteristics of the object and partially learns the irrelevant details of the images, which leads to overfitting problems. To address the overfitting problems, we use different cameras in the process of making the homemade dataset to capture images of workpieces in several industrial scenarios.

For a new network architecture or a new semantic segmentation task, it is necessary to collect and annotate thousands of images for training a high-performance deep neural network from scratch. While the accuracy and the fault tolerance of the trained model are higher with more diverse and representative image data, the amount of work involved in collecting and annotating the images is too large for many researchers to complete. Therefore, we utilize transfer learning technology to address this problem. Because most of the features of the images or the tasks are related, we can share the pretrained model parameters to the new network model via transfer learning, which greatly quickens and optimizes the learning efficiency and the effectiveness of the new model. In this research, the object we perform semantic segmentation on is a new workpiece. Since ImageNet for training VGG16 contains a wide variety of images, the workpieces are related to the images in ImageNet in terms of color, shape, and other features. Therefore, we utilize transfer learning to train a semantic segmentation neural network for binary classification based on the pretraining parameters of the VGG16 classification model. According to the PASCAL VOC 2012 standard [42], we manually annotate the dataset of the workpieces for different industrial scenarios and augment the dataset with data augmentation techniques (including image rotation, crop, and horizontal and vertical flip). The homemade dataset consists of five categories (including the background) and contains a total of 950 images. Based on the experience of dividing data sets for training neural networks [43], we manually divide the entire dataset into three parts: the training set, the validation set, and the test set. We put 80% of the images into the training set, 10% to run as a validation set during training, and then use the remaining 10% as a test set to test the inference performance of the trained model on the task scenes images. In order to ensure the uniformity and the diversity of the image distribution in each data set used for training and testing, the three data sets all contain workpiece images in multiple different scenes, which can improve the accuracy of recognition.

Based on the original FCN architecture, we propose a modified FCN architecture, FCN-atrous-2s, by fusing shallower image features and atrous convolution. According to the methods and the procedures of FCN training, we copy the model parameters from the VGG16 classification network via transfer learning to initialize the weights and the biases of FCN-atrous-2s, and then train our modified FCN from the coarse to fine level. To compare the performance between our model and the other state-of-the-art models, we also trained SegNet [44] and FCN-8s on the homemade dataset. In [14], Wong et al. presented SegICP, which couples SegNet and multi-hypothesis ICP to achieve both robust pixel-wise semantic segmentation as well as accurate and real-time 6-DOF pose estimation for relevant objects. The segmentation results of the FCN-atrous-2s, SegNet, and FCN-8s are shown in Figure 5. More specifically, we utilize four different quantitative metrics to estimate the precision of the semantic segmentation, as seen in Table 1: pixel accuracy, mean accuracy, mean IOU (mean intersection-over-union), and f.w. IOU (frequency weighted IOU). Because the majority of pixels are background pixels, pixel accuracy is not the primary reference criterion. Instead, the mean IOU is a more reasonable evaluation criterion for semantic segmentation tasks. According to the results in Table 1, the mean IOU of our approach is higher than the SegNet and FCN-8s’ by two percent and five percent, respectively.

Our modified FCN models for semantic segmentation are trained and tested with Caffe [45], a deep learning framework made with expression, speed, and modularity in mind. Our model is trained using a computer with an Intel Core i7-4800MQ 2.7 GHz and an NVIDIA GTX 970 for CUDA (Compute Unified Device Architecture) parallel computations. Training our models takes up to 32 h after 10,000 iteration steps. We also trained SegNet and FCN-8s on our homemade dataset with Caffe. The run-time speeds per component are as follows: 0.2 s per forward inference of the FCN-atrous-2s, 0.3 s of the SegNet, and 0.3 s of the FCN-8s.

### 3.2. Pose Error Evaluation Method Based on Robot Vision System

Although in the visual window, the quality of the point cloud registration can be visually and intuitively judged. However, in order to exactly quantify the accuracy of registration, we propose a method to evaluate the accuracy of the transformation matrix based on the robot vision system. As shown in Figure 6, the calibration target and workpieces are attached to the end effector of the industrial robot, and the two can be considered to constitute a rigid body. One of the characteristics of a rigid body is that when different parts of the rigid body are transformed between different poses with respect to the specified frame, the description of the transformation matrix remains the same; that is, when we determine the transformation matrix of a part of the rigid body in a specified frame, we can correspondingly obtain the transformation matrix of the complete rigid body between the two poses.

Because the end-effector and the workpiece constitute a rigid body, according to the above ideas, we can obtain the following expression:(12)$$T_{ee}=T_{reg}$$
where $T_{reg}$ is the transformation matrix of the workpiece between the two arbitrary poses, and $T_{ee}$ is the transformation matrix of the end-effector between the two arbitrary poses in the camera frame. $T_{ee}$ can be solved not only by Equation (12) but also by the hand-eye calibration principle of the robot and the camera. The hand-eye calibration diagram is shown in Figure 7. Assuming that the industrial robot is in two poses, Pose_targrt and Pose_source, the 4 × 4 transformation matrix *C_tgt_* and *C_src_* of the end-effector frame E relative to the camera frame C and the relationship between the *C_tgt_* and *C_srct_* can be expressed as:(13)Ctgt=TCB−1TEBtgt and Csrc=TCB−1TEBsrc
(14)$$C_{tgt}=T_{ee}C_{src}=T_{reg}C_{src}$$
where TEBtgt and TEBsrc are 4 × 4 transformation matrices of the end-effector frame E with respect to the robot base frame B in the two poses, which can be calculated using the forward kinematics of the robot. TCB−1 is the inverse matrix of TCB, and TCB, which is figured out by hand-eye calibration, is the transformation matrix of the camera frame with respect to the robot base frame.

In theory, Equations (12) and (14) are correct, but $T_{reg}$ and $T_{ee}$ cannot be exactly equal due to the accuracy of the point cloud registration and the robot motion. Therefore, in the absence of expensive measuring equipment, we use hand-eye calibration to evaluate the error of the point cloud registration, as mentioned above. Although industrial robots have errors during movement, they belong to the submillimeter level and are smaller than the registration error.

Therefore, in order to evaluate the accuracy of the point cloud registration, Equation (14) is rewritten as Equation (15), thus the error $\varepsilon_{T}$ is converted to compare the difference between the matrices $C_{tgt}{}^{\prime }$ and *C_src_*.
(15)$$C_{tgt}{}^{\prime }=T_{reg}C_{src}$$
(16)$$\varepsilon_{T}=\left\{C_{tgt}-T_{reg}C_{src}\right\}=\left\{C_{tgt}-C_{tgt}{}^{\prime }\right\}$$
where the matrices $C_{tgt}$ and $C_{tgt}{}^{\prime }$ contain 3 × 3 rotation transformation matrices $R_{tgt}$, $R_{tgt}{}^{\prime }$ and 3 × 1 translation transformation vectors $p_{tgt}$, $p_{tgt}^{\prime }$.

For calculating the error between two matrices, the elementwise subtraction between the matrices does not directly reflect the magnitude of the error, thus we evaluate the errors of the rotation matrix R and the translation vector $p_{t}$. The rotation matrices $R_{tgt}$ and $R_{tgt}{}^{\prime }$ should theoretically be equal, and $R_{tgt}{\left(R_{tgt}{}^{\prime }\right)}^{-1}$ should be equal to the identity matrix $I_{3\times 3}$. Thus, the error of the two rotation matrices can be converted into an error between $R_{tgt}{\left(R_{tgt}{}^{\prime }\right)}^{-1}$ and the identity matrix $I_{3\times 3}$. As shown in the following Equation (17), we convert the rotation matrix into a rotation vector via the Rodrigues formula, and then solve the 2-norm of the vector. The smaller the value of the 2-norm, the better the registration accuracy.
(17)$$\varepsilon_{R}={\left\|Rodrigues(R_{tgt}{\left(R_{tgt}{}^{\prime }\right)}^{-1})\right\|}_{2}$$

Similarly, the errors of the translation vectors $p_{tgt}$ and $p_{tgt}^{\prime }$ can also be calculated as by a 2-norm as follows:(18)$$\varepsilon_{p}={\left\|p_{tgt}-p_{tgt}^{\prime }\right\|}_{2}$$

### 3.3. Results of Pose Estimation

#### 3.3.1. Parameters Evaluation of Preprocessing Phase

To improve the efficiency of the point cloud registration and meet the real-time requirements of industrial setting applications, in the preprocessing phase, we utilize the statistical outlier removal and the VoxelGrid method to downsample the point cloud, which effectively improve the calculation speed and reduce the point cloud registration time, and for the case of multiple same objects in the one scenario, the Euclidean distance clustering method is used to further segment the independent point cloud of the workpiece.

First, we study the effect of the parameter K of the statistical outlier removal method on the removal of outliers. As described in Section 2.2.1, we first calculate the mean distance of each point to the K nearest neighbors, then the mean and the standard deviation of Gaussian distribution, formed by the mean distance result of the entire point cloud, are used to determine the attribute of each point—outliers or not. Therefore, we use the control variable method to evaluate the processing time and the number of outliers removed by changing the value of K, as shown in Figure 8. By comprehensively comparing the evaluation results of the statistical outlier removal method, we set the K value as 70, the average number of outliers removed is 261, and the running time is 0.398 s. The effect of outliers removal is shown in Figure 9.

Then, we utilize the VoxelGrid method to downsample the point cloud after outliers removal. During the experiment, the size of voxel grid *Ls*, which is an important variable that affects the efficiency of downsampling and the accuracy of point cloud registration, should be set first, thus we design an experiment to evaluate the time cost and the effect of *Ls* on the accuracy of pose estimation, which is performed by utilizing ICP. The evaluation results are shown in Figure 10, the errors of pose estimation become larger as the voxel grid size *Ls* increase, while the runtime is reversed. Then, we comprehensively analyze the experimental results and set *Ls* as 0.008m, and the mean runtime of point cloud downsampling is 0.185s.

For the case of multiple same workpieces in the one scenario, we employ the Euclidean distance clustering method, in which *d_th_* is the key parameters, to extract the independent point cloud of the workpiece. After validating the performance of the method on the corresponding point clouds and taking the actual pose of the workpieces into account, we finally set *d_th_* as 0.01 m, with which the success rate is above 97%. However, there are some special cases, such as two of the same workpieces overlapping, as shown in Figure 11, we cannot address this problem for the time being. We focus on solving it in the next step of the study by using instance segmentation technology.

In summary, the parameter settings in the preprocessing phase are shown in Table 2.

#### 3.3.2. Results Evaluation of Pose Estimation

To meet the requirements of the hand-eye calibration between the robot and the camera, in the experimental phase, we first randomly move the industrial robot with the calibration target to 21 different poses, then we figure out the constant transformation matrix TCB−1 of the robot base relative to the RGB camera of the RGB-D sensor, and save the results [the translation vector (0.396, 0.121, 1.005), quaternion (0.062, 0.708, 0.703, −0.014)] for the calculation of subsequent pose estimation.

To evaluate the errors of pose estimation, the workpieces (including four types of workpieces) are mounted on the end-effector of the industrial robot, as shown in Figure 6b. Then, we randomly move the industrial robot with the workpieces to 41 different poses. In order to guarantee that the distance between each new pose and the initial pose (the first pose) is increasing, we set the increment by 5 mm (the resultant displacement of X/Y/Z). Correspondingly, the rotation angle about the X/Y/Z axis of the robot base frame is increased by two or three degrees.

In the process of robot movement, the RGB-D sensor captures the RGB and the depth images of the workpieces in each pose, and the computer synchronously records the joint angle $\vec{\theta }$, which is used to calculate the corresponding forward kinematic transformation matrix TEBi of the industrial robot in the current pose. Then, we use the FCN-atrous-2s proposed in this paper to segment the workpiece in the color map, obtaining the point cloud of the workpieces under each pose with depth information. To unify the evaluation benchmark, we set the workpiece point clouds in the first pose as the initial point cloud $PC_{pose1}$, with the point clouds in the next 40 poses all using $PC_{pose1}$ as the target point cloud. We calculate the transformation matrix $T_{reg}$ of the point cloud of these poses relative to $PC_{pose1}$ by applying the point cloud registration algorithm. We substitute the experimental data into Equations (17) and (18) to calculate the rotation matrix error $\varepsilon_{R}$ and the translation vector error $\varepsilon_{p}$. To compare the effect of pose estimation, we reproduce the FPFH-ICP [46] and Go-ICP [47] registration algorithm and process 41 experimental data samples to compare their results with that of the algorithm presented in this paper, as shown in Figure 12. In order to compare the effects of different neural networks on the pose estimation results, we also use SegNet trained on our homemade dataset to perform semantic segmentation, then we apply the ICP method to estimate the pose of workpieces. To unify the evaluation benchmark, we also utilize the above 41 poses to evaluate the errors of pose estimation, as shown in Figure 13a,b.

As shown in Figure 12 and Figure 13, the evaluation errors of ICPs increase with distance increasing, because the changes in distances and rotation angles result in decreasing the number of points participating in the registration, i.e., the overlap area decreases, which directly affects the point cloud registration effect [48]. Additionally, for the error mutation problem in Figure 12 and Figure 13, we use the control variable method to fix the preprocessing and the registration configuration parameters, and process and compare multiple sets of point cloud data. We find that some of the source point clouds and the target point cloud themselves have large differences in their poses. In addition, the source point cloud in these poses contains a large number of scattered points, as shown in Figure 14. By the analysis, we can see that there are two main reasons for the emergence of a large number of scattered points: (1) the calibration problem between the RGB camera and the depth camera of the RGB-D sensor, which is a Kinect v2 camera developed by Microsoft Corporation. Specifically, the RGB camera and the depth camera are not coincidental in terms of the physical structure. Before their first use, the two need to be calibrated to the same coordinate system. This calibration process will introduce errors that result in the pixels of the RGB image and the depth image being unable to have a one-to-one correspondence, thus it is easy to generate scattered points at the edge contour; (2) the accuracy of the FCN-atrous-2s and the SegNet on the semantic segmentation of the RGB images. Although the FCN has made great progress in semantic segmentation, there are still differences between the results of the semantic segmentation and the Label, which lead to the results of the segmentation not completely coinciding with the contour of the actual object, thus it is possible to exceed the contour of the actual object at the edge. Therefore, it is easy to mistakenly incorporate some point clouds of other objects into the workpiece point cloud scene. The existence of these other objects’ point clouds greatly affects the registration accuracy of the corresponding point clouds.

We calculated the mean, the standard deviation, and the runtime of the rotation error and the translation error of four types of workpieces. The computer configuration is the same as that used to train the FCN-atrous-2s and SegNet. As shown in Table 3, the errors of the three point cloud registration methods are close, but in terms of the runtime, the calculation of the normal features, the point feature histogram, and the rough registration process take an average of 38.493 s, which ultimately results in a much longer runtime for the FPFH-ICP method than for that proposed by this work. Similarly, Go-ICP takes an average of 20.7 s to build distance transform. By comparing the results of this paper to ICP and SegNet-ICP, it can be proven that the performance of the neural network has an effect on the pose estimation results. After a comprehensive comparison, our method, which couples FCN-artrous-2s and ICP to perform pose estimation for the workpieces in industrial scenario, is more suitable for our grasping system.

However, considering the real-time requirements of industrial robots for grasping tasks, the method of this paper is not practical to a certain extent. Therefore, we intend to introduce CUDA parallel computing to quicken the registration process of the point cloud and reduce the runtime to subseconds.

## 4. Conclusions

In this work, we present an efficient, practical workpiece semantic segmentation and pose estimation method using RGB-D data. These data are used for semantic segmentation and depth data for the pose estimation, and our solution has a rotation error < 2 degrees (0.035 rad) and a translation error < 10 mm. Based on the original FCN, we fuse the shallower features of the network with atrous convolution and train the FCN-atrous-2s on the homemade dataset by utilizing transfer learning to achieve dense predictions for the RGB images. Then, we determine the transformation matrix of the source point cloud and the target point cloud by using the point cloud registration method. To evaluate the accuracy of the transformation matrix, we propose a practical pose error evaluation method based on the robot vision system. The advantage of this method is that accurate evaluation results can be obtained without relying on expensive measuring equipment. Although the performance of the proposed method in this paper is not lower than the method described in the reference, there are still some problems in the algorithm implementation of image segmentation and point cloud registration, thus we have not yet obtained the optimal solution. We are currently working towards improving the performance of the pose estimation algorithm architecture by extending the workpieces dataset to contain more industrial workpieces and addressing the problem of simultaneously identifying multiple workpieces. Through the introduction of a CUDA parallel computing method throughout the entire system, we are working hard to improve the running speed of the entire process in order to estimate the pose of the workpieces in real time.

## Figures and Tables

**Figure 1 sensors-19-01873-f001:**
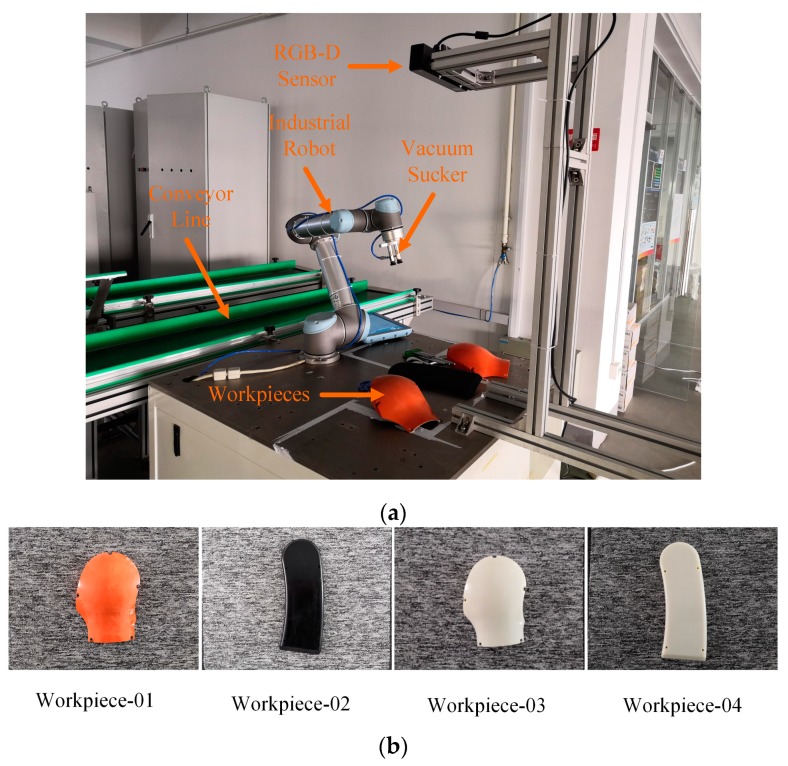
The experiment platform and workpiece types. (**a**) The workpiece grasping experiment platform, the vacuum sucker is used to grasp the workpiece from the platform onto the assembly line. (**b**) The four different types of workpieces that vary in shape and appearance.

**Figure 2 sensors-19-01873-f002:**
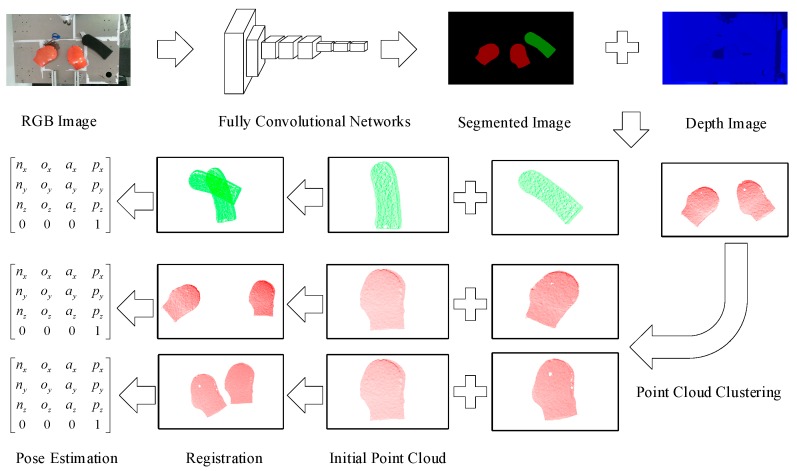
Pose estimation scheme. We first use an RGB-D (color and depth) sensor to capture the RGB and depth images of scattered workpieces, then the pretrained modified Fully Convolutional Network (FCN) takes the RGB image as the input and outputs a pixelwise classified segmented image, including the shape of the workpieces (filled with different colors) and the background (filled with black), with the same dimensions as the input image. The segmented point clouds of the workpieces can be acquired easily by mapping them from the two-dimensional (2D) pixel sets of the workpieces, which are extracted from the segmented image, to the corresponding depth image. We utilize the point cloud registration algorithm to determine the transformation matrix between the segmented point cloud and the initial point cloud. Because the pose of the initial point cloud relative to the base frame of the industrial robot is known, the pose of the segmented point cloud is therefore easy to calculate via a matrix transformation. In particular, when there are multiple workpieces of the same type in the same scenario, we further segment the point cloud using Euclidean distance clustering.

**Figure 3 sensors-19-01873-f003:**
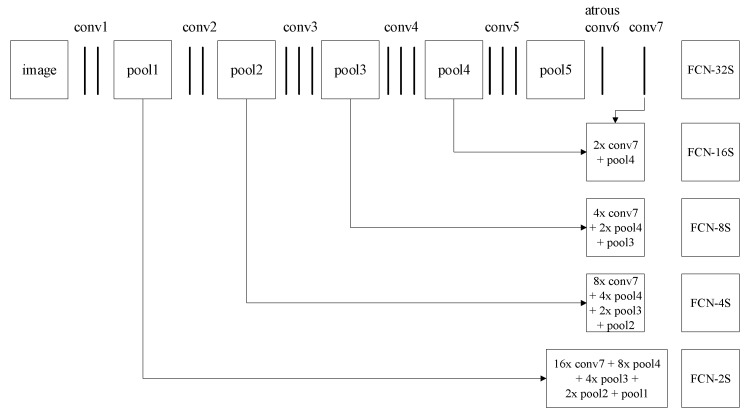
Multiscale feature fusion. We first change the VGG-based FCN’s convolution six layer, which has the highest parameter calculation quantity, to atrous convolution with a kernel size of 3 × 3. The atrous convolution can increase the receptive fields without reducing the spatial dimension while simultaneously decreasing the computational complexity and inference time. Then, based on the FCN-8s, we use the skip architecture to fuse the shallow detail features in the neural network and train the FCN-atrous-2s model on the homemade dataset.

**Figure 4 sensors-19-01873-f004:**
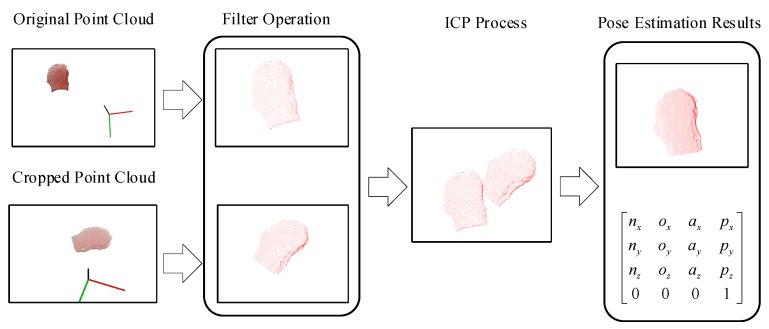
Pose estimation schematic. In the front-end loading process, the overlap ratio between the surfaces of each workpiece can be guaranteed for the robot grasping task, which provides sufficient initial conditions for the point cloud registration. We first preprocess the segmented and the initial point clouds by removing the scattered points and downsampling to reduce the amount of point cloud data while keeping the surface features of the workpieces unchanged. Then, we use the Iterative Closest Point (ICP) method to minimize the registration error and output the rigid transformation matrix between the two point clouds.

**Figure 5 sensors-19-01873-f005:**
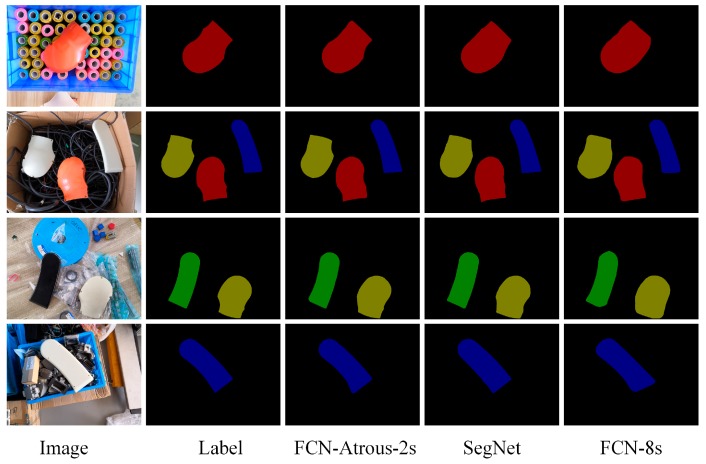
Comparison of semantic segmentation results. The segmentation results of the FCN-8s and Segnet are smoother to cover the original shape features in places where the object boundary changes are greater than FCN-atrous-2s. In comparison, the segmentation results of the FCN-atrous-2s are more elaborate, which shows the variation characteristics of the boundary and which are closer to the result of the Label.

**Figure 6 sensors-19-01873-f006:**
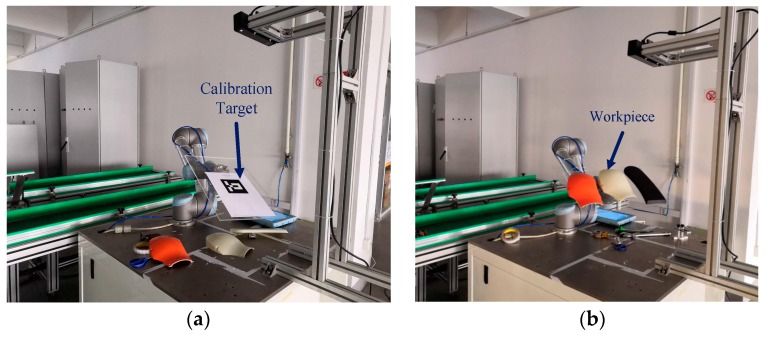
Hand-eye calibration and pose estimation experiment platform. (**a**) The camera is mounted outside the industrial robot, and the relative position between the robot and the camera can be represented by a constant matrix. We made a calibration target, namely, the Aruco Marker, which is flat on the calibration target. (**b**) The workpieces are mounted on the end-effector of industrial robot, and the two constitute a rigid body.

**Figure 7 sensors-19-01873-f007:**
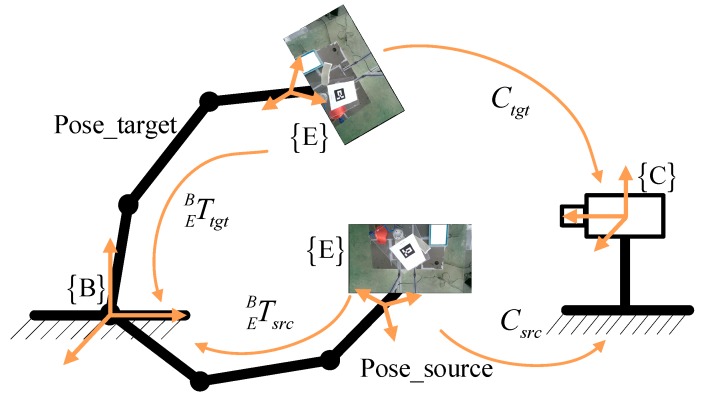
Hand-eye calibration diagram. The scenario in this paper belongs to the hand-eye calibration problem, where the camera and the robot base are relatively fixed.

**Figure 8 sensors-19-01873-f008:**
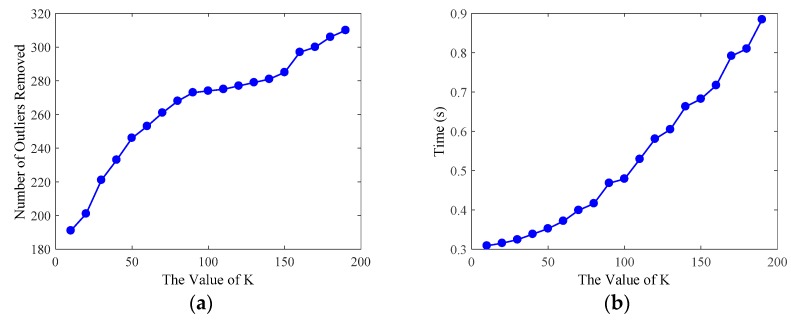
Evaluation results of the statistical outlier removal method. We set the number K of each point’s nearest neighbors as the abscissa, and the evaluation was conducted on 20 point clouds using a sequence of different K values. (**a**) The number of outliers removed by statistical outlier removal method with respect to different K values. (**b**) The runtime of the method. Both the number of outliers removed and runtime increase with an increase of K value.

**Figure 9 sensors-19-01873-f009:**
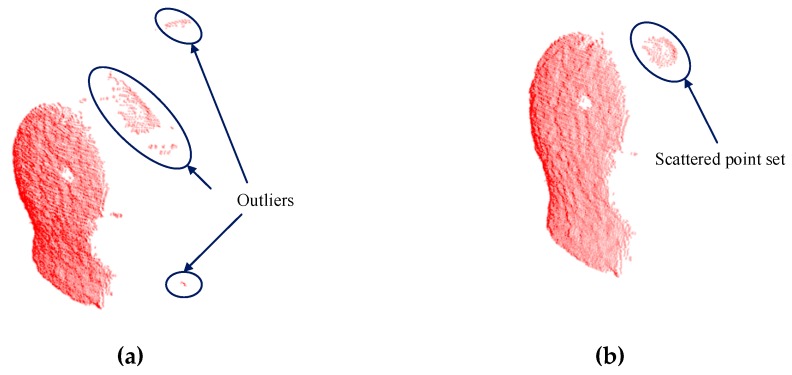
Comparison of outliers removal results. (**a**) There are a lot of obvious outliers in the original point cloud before using outliers removal. (**b**) A majority of the outliers are removed after using outliers removal, but there are some very dense point sets that cannot be removed totally.

**Figure 10 sensors-19-01873-f010:**
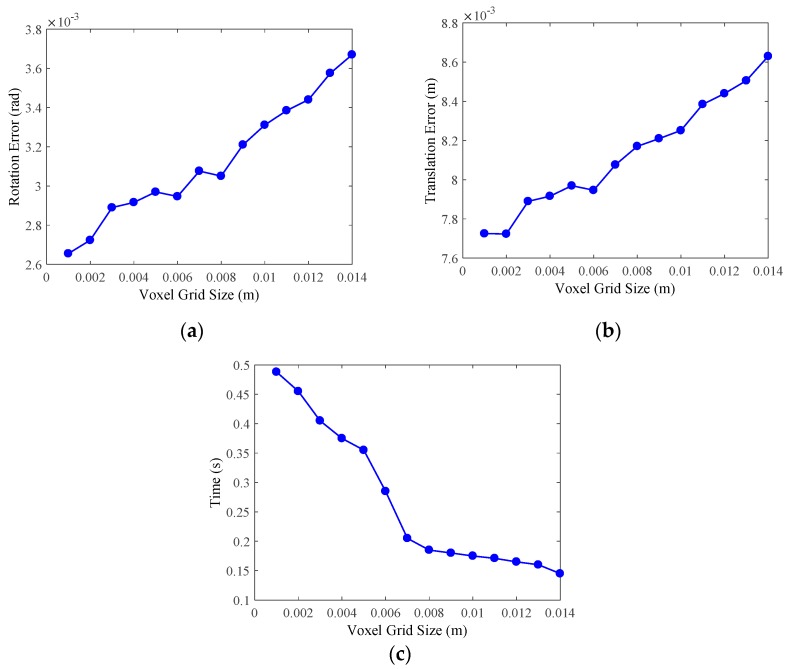
Evaluation results of the VoxelGrid downsampling method. We set the voxel grid size as the abscissa, and the evaluation was conducted on 20 point clouds that are processed by the statistical outlier removal method. (**a**) The rotation matrix errors of point cloud registration by using ICP. (**b**) The translation errors of point cloud registration by using ICP. (**c**) The runtime of VoxelGrid downsampling.

**Figure 11 sensors-19-01873-f011:**
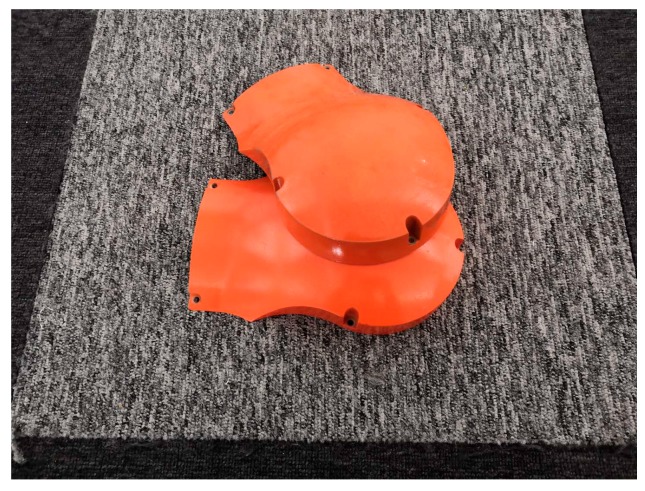
Two of the same workpieces overlapping, resulting in difficulty segmenting the point clouds in three-dimensional (3D) space.

**Figure 12 sensors-19-01873-f012:**
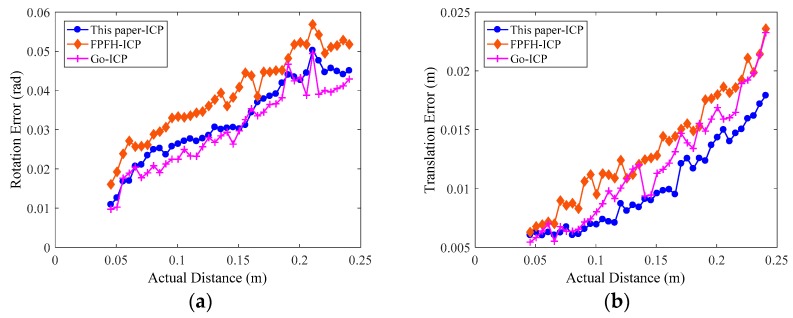
Errors of the pose estimation with the semantic segmentation results of FCN-atrous-2s. We set the actual Euclidean distance of the robot end-effector in the corresponding two different poses as the abscissa. (**a**) The rotation matrix errors obtained using the three registration methods fluctuate with an increasing distance. (**b**) The translation errors increase with an increase in the actual distance, but their fluctuation is also relatively large.

**Figure 13 sensors-19-01873-f013:**
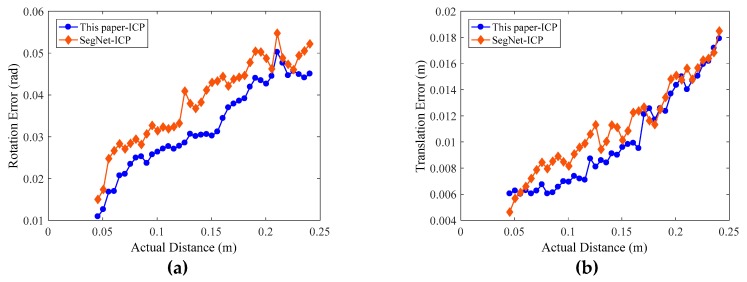
Errors of the pose estimation with FCN-artous-2s and SegNet, respectively. We set the actual Euclidean distance of the robot end-effector in the corresponding two different poses as the abscissa. (**a**) The rotation matrix errors. (**b**) The translation errors. The errors are obtained by using two semantic segmentation method and also increase with an increase in the actual distance.

**Figure 14 sensors-19-01873-f014:**
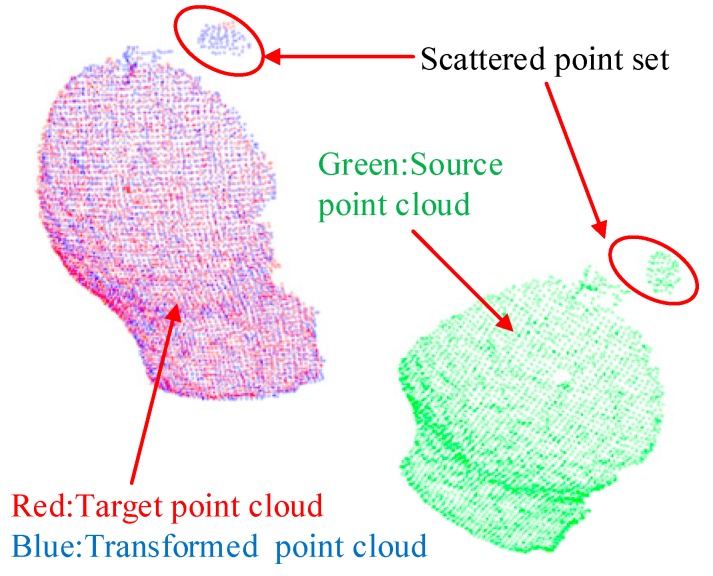
Typical phenomenon of the scattered point cloud. These scattered points are not removed by the statistical outlier removal method in the preprocessing phase, mainly because the points in the scattered point set are very dense and cannot be removed at one time by the algorithm.

**Table 1 sensors-19-01873-t001:** The performance of the Neural Networks for Semantic Segmentation

Neural Nets	Pixel Accuracy	Mean Accuracy	Mean IOU	f.w. IOU	Inference Time (s)
FCN-8s	0.9524	0.9182	0.8353	0.9128	0.3
SegNet	-	0.9289	0.8664	-	0.3
FCN-atrous-2s (ours)	0.9673	0.9471	0.8857	0.9495	0.2

* IOU = intersection-over-union, f.w. IOU = frequency weighted IOU.

**Table 2 sensors-19-01873-t002:** The key parameters in the preprocessing phase.

Method	Statistical Outlier Removal	VoxelGrid	Euclidean Distance Clustering
	K	*Ls* (m)	*d_th_* (m)
Value	70	0.008	0.01

**Table 3 sensors-19-01873-t003:** The performance of pose estimation methods.

Method	Rotation Error (rad)		Translation Error (m)		Runtime (s)
	Mean	Std	Mean	Std	
**This paper-ICP**	0.032	0.010	0.010	0.004	2.26
**FPFH-ICP**	0.039	0.011	0.014	0.004	42.38
**Go-ICP**	0.030	0.010	0.012	0.003	23.53
**SegNet-ICP**	0.038	0.009	0.011	0.005	2.47

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
