# Peer review of "RGB-D-Based Pose Estimation of Workpieces with Semantic Segmentation and Point Cloud Registration"

_sensors, 2019, doi:10.3390/s19081873_

Round 1

Reviewer 1 Report

This paper presents a RGB-D based object pose estimation for industrial tasks. In the proposed method, the point cloud of the target 3D object is extracted by the segmentation, and the object pose is estimated by the ICP. 

In this paper, the technical novel contribution is quite weak:

- The pose estimation is basically done by the conventional ICP with some preprocessing for the speed-up, which is not evaluated enough (e.g., voxel size).

- The accuracy of the pose highly depends on segmentation results (as also mentioned in this paper), so that its analysis (like sensitivity) should be sufficiently evaluated and discussed.

- The proposed hand-eye calibration is a common approach for comparative evaluations when a robot arm is available. 

- The proposed method is not evaluated with the state-of-the-art pose estimation methods and also with other segmentation methods.

- The segmentation is based on FCN with some modification, but the basic idea is not novel.

- In this paper, the segmentation results show only single target object, and there is no semantic aspect. In industrial environments, the backgrounds are highly cluttered with a variety of objects, but this paper does not handle such issues, which are more challenging on segmentation and critically affect the pose estimation in the proposed method.

Overall, there are many recent relevant works, but they are not well-studied and not compared, e.g., 

- SegICP: Integrated deep semantic segmentation and pose estimation (2017)

- PoseCNN: A Convolutional Neural Network for 6D Object Pose Estimation in Cluttered Scenes (2017)

- Visual Object Recognition and Pose Estimation Based on a Deep Semantic Segmentation Network (2018)

- Deep Object Pose Estimation for Semantic Robotic Grasping of Household Objects (2018)

and etc….

I cannot see the proposed method is a better approach than other RGB-D based object pose estimation methods which have no such segmentation step (but fully support fast and robust performance) or ones with recent CNN-based methods. Please clarify and evaluate this point. 

Minor:

- In equation(8): q_0,1,2,3 --> q_w,x,y,z

- line 377: the translation vector q --> p

- In reference: missing publication years

Author Response

Dear reviewer:

We wish to thank you for the time and effort you have spent reviewing our manuscript. Motivated by your comments, we have deeply reconsidered the integrality of our experiments and tried to fix all the problems you pointed out. Please refer to the attached responses. Thank you again.

Reviewer 2 Report

The paper presents a pose estimation approach combining a deep learning approach for object detection and labeling of the pixels belonging to the objects of interest. The point cloud corresponding to the labeled "object" pixels is then used for estimating the pose of the object with respect to an "original" pose (the term "original" is not defined clearly in the paper and the reader needs to wait until the section on experimental results to find out that it is only the initial position of the object on a series of 21 images taken from different points of view from a still RGB-D camera (actually a Kinect V2 sensor)). The paper says little about the types of objects of interest except that they can be found in "industrial environments". Only one object is shown in the paper and it is not described (what is this uniformly red object by the way?). The geometry of the object is very simple and its appearance is even more simple (uniform red color all over the surface). The reader wonders why a CNN is even needed to detect such an object since a simple analysis of the hue of the HSV image would have probably been sufficient.

The literature review on object detection and "tracking / pose estimation" is not complete and important references are not covered. There are too many missing references to list here.

The description of the proposed learning architecture is a bit confusing and difficult to follow. The advantages of the selected architecture are described satisfactorily though.

The pose estimation approach based on the alignment of point clouds of the object is basically a straight implementation of the well-known ICP algorithm. ICP requires that the pose difference between views must be small (with a good percentage of overlap between views) for the algorithm to converge. Consequently, the rationale for using such an approach in an industrial context is not clear since it is very unlikely that, in an object grasping application (for scattering objects on a treadmill as mentioned in the paper), such close initial poses will be met. Some approaches such as GoICP, which exploits Branch-and-Bound optimization, can circumvent this limitation and converge to the global minimum even though the initial poses are not close. Through its hyper parameters, GoICP can eliminate the need for cleaning up the initial point clouds through the setting of a trimming factor.

The theory for computing the pose is OK and is based on classical SVD procedure.

The main weakness of the paper is that the proposed approach is tested on a home-made dataset that does not allow comparison with other methods found in the literature. The paper is silent on which objects are used for training: what is their geometry, surface texture, surface reflectance, lighting conditions in the workspace, etc.). As such, it is difficult for the reader to figure out how can the proposed system generalize to a real industrial setting. It is mentioned that industrial environments are structured and, consequently, are easier to deal with than other types of environments (outdoor scenes for instance). This is partially true since industrial workspaces are corrupted by shadows, lighting, specular reflections, occlusion between objects, etc. The robustness of the proposed approach is not discussed or even alluded to in the paper. It is not mentioned either if the network can generalize or not to objects that have never seen (even ion training). The number of false alarms and non detection is not given in the paper. Have experiments been conducted to address these issues?

The experimental results are not convincing since an error of 2 degrees in rotation and 10 mm in translation are achieved by the system. This is huge. Current dynamic pose estimation systems using deep learning on RGB information alone achieve much better results on a large variety of objects with complex geometry and surface texture. It is mentioned that a source of error comes from the different point of view between the RGB camera and the range camera of the Kinect V2. It is strange since the extrinsic calibration parameters between the two sensors is available (or, at least, the mapping matrix between them). Was this information used for the development? The pose estimation results were compared to the ones obtained with an AruCo marker imaged with the same sensor than the one used for the experiments. This is OK, but it would have been much more convincing if a more accurate camera would have been used for collecting the ground truth poses. In Fig. 7 (a), it is surprising that the error in the estimation of rotation seems to decrease with distance...to say the least, this is very counterintuitive.

Although the runtime (for the single object presented in the paper) is low compared to a competing method (3 sec vs 46 sec), it is still far from real-time performance.

The paper cannot be accepted in its current form.

Author Response

(The authors gave the same response as above.)

Round 2

Reviewer 1 Report

This paper is properly revised with several improvements. The approach is not novel enough, but its position is reasonably clarified and acceptable for industrial applications.

There are some minor points for the final acceptance:

1. In figure 12-13: Both errors seems to be linearly increased (actually, they looks like linearly accumulated even thought they do not) by the actual distances. In this revised paper, it is shortly mentioned that it is due to large differences between both poses (line 557-558). The authors' response (answer about Point 7 of Reviwer 2) also explains that it relies on the capability of the ICP method. Even though I understand these issues, now I wonder what makes the errors linearly increasing from random poses and how to get random poses for these evaluations.

2. For such evaluations, I think it might be better to use spherical reflective markers, which provides more accurate ground truth data than what visual makers (pattern) do.

3. Equation 14/15/16: C_1 = T_ee C_2 = T_reg C_2 (?) ---> C_1 = C_2 T_ee = C_2 T_reg (row-major)

Author Response

Dear reviewer:

We wish to thank you very much for the time and effort you have spent reviewing our manuscript. Motivated by your comments, we have deeply reconsidered the integrity of our experiments and tried to fix all the problems you pointed out. Please refer to the attached responses. Thank you again.

Reviewer 2 Report

Revision covers comments in the first review.

The authors should proofread the manuscript a last time for typos

Author Response

(The authors gave the same response as above.)
